# CAT:
# POST-TRAINING QUANTIZATION ERROR REDUCTION VIA CLUSTER-BASED AFFINE TRANSFORMATION

## ABSTRACT

Post-Training Quantization (PTQ) reduces the memory footprint and computational overhead of deep neural networks by converting full-precision (FP) values into quantized and compressed data types. While PTQ is more cost-efficient than Quantization-Aware Training (QAT), it is highly susceptible to accuracy degradation under a low-bit quantization (LQ) regime (e.g., 2-bit and 4-bit). Affine transformation is a classical technique used to reduce the discrepancy between the information processed by a quantized model and that processed by its full-precision counterpart; however, we find that using plain affine transformation, which applies a uniform affine parameter set for all outputs, is ineffective in low-bit PTQ. To address this, we propose Cluster-based Affine Transformation (CAT), an error reduction framework that applies cluster-specific affine transformation to align LQ and FP outputs. CAT directly refines quantized outputs with only a negligible number of additional parameters. Experiments on ImageNet-1K demonstrate that CAT consistently outperforms prior PTQ methods across diverse architectures and low-bit settings, achieving up to 53.18% Top-1 accuracy on W2A2 ResNet-18, and delivering improvements of more than 3% when combined with strong PTQ baselines. We plan to release CAT's code alongside the publication of this paper.

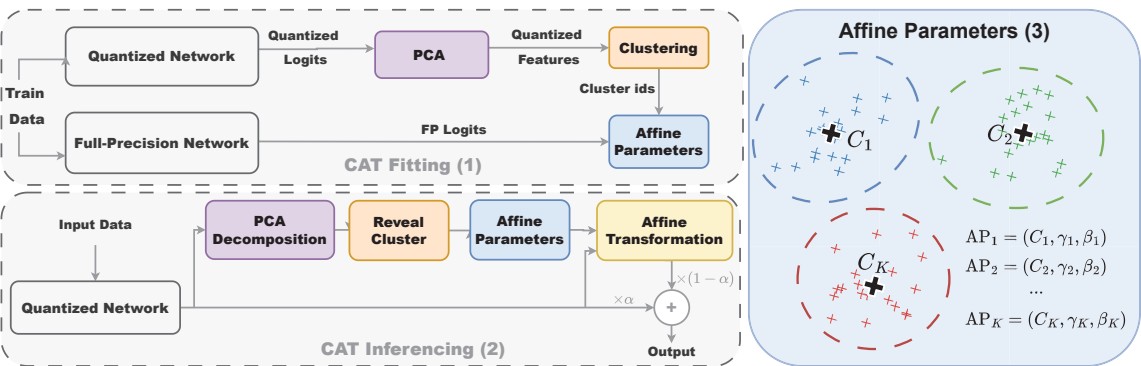

Figure 1: Cluster-based Affine Transformation (CAT); (1): CAT Fitting is the process of fitting the clustering model and estimating $\gamma$ and $\beta$ parameters for each cluster.(2) CAT Inferencing, the process of error reduction using CAT. (3) Affine Parameters (AP), a set of parameters we store for each cluster of CAT.

# 1 INTRODUCTION

Deep neural networks (DNNs) achieve remarkable performance on different computer vision tasks Ravi et al. (2025); Xiao et al. (2025); Zhu et al. (2025); Li et al. (2025b); Wang et al. (2025); Zanella et al. (2024). but demand prohibitive memory and computation due to millions of floating-point parameters. Quantization, which reduces the precision of weights and activations to low bit-widths, offers an efficient compression strategy and is now widely supported by modern hardware. Post-Training Quantization (PTQ) is a practical model compression which determines quantization parameters without retraining or fine-tuning, requiring only a small calibration set. Despite this promise, PTQ suffers from a severe accuracy degradation when pushed to LQ regimes. In this work, we investigate the affine transformation Weisstein (2004); Ma et al. (2024) ability to restore PTQ output errors to refine its accuracy degradation.

We first investigate a uniform set of affine transformation parameters, referred to as a plain affine transformation. As shown in Table 1, this approach fails to recover predictions and often further degrades the Top-1 accuracy of LQ ResNet-18. To alleviate the issue, we leverage clustering PTQ outputs and find a specific affine transformation parameter set for each cluster. Our proposed method, Cluster-based Affine Transformation (CAT), yields superior error reduction results for PTQ and especially the low-bit quantization (LQ) (e.g., 2-bit) regime. This strategy substantially reduces the output gap between FP and LQ, improving accuracy under low-bit settings with only a negligible number of additional affine parameters. Compared to the baseline, CAT raises the Top-1 accuracy of 2-bit quantized (W2A2) ResNet-18 to 53.18%. For compact DNNs such as MNasX2, CAT achieves a $+1\%$ accuracy improvement in the 2-bit quantization of both weight (W) and activation (A) (W2A2). Because of its ability to directly restore output errors, CAT can be used as a plug-in for a wide range of

| $w$ | $a$ | Method | Acc (%) |
|---|---|---|---|
| 2 | 2 | No affine transformation | 52.84 |
| | | Plain affine transformation | 52.32 |
| | | CAT (Ours) | **53.18** |
| 4 | 2 | No affine transformation | 58.58 |
| | | Plain affine transformation | 58.22 |
| | | CAT (Ours) | **58.80** |
| 2 | 4 | No affine transformation | 65.12 |
| | | Plain affine transformation | 65.17 |
| | | CAT (Ours) | **65.25** |
| 4 | 4 | No affine transformation | 69.17 |
| | | Plain affine transformation | 69.14 |
| | | CAT (Ours) | **69.27** |

Table 1: Top-1 accuracy of ResNet-18 under quantization: comparison between no affine transformation, plain affine transformation, and CAT (ours).

PTQ methods. Our results show that, with this capability, CAT improves Top-1 accuracy by more than 3% for some PTQ methods. Our contributions can be summarized as follows:

- We propose Cluster-Affine Transformation (CAT), a novel output-level error reduction method that leverages the natural clusterability of logits to improve alignment.

- We introduce a novel state-of-the-art post-training quantization framework, which achieves consistent accuracy improvements across diverse architectures and LQ settings, surpassing prior PTQ methods with negligible overhead.

- We achieve higher Top-1 accuracy compare to PTQ baselines on ImageNet-1k for different DNNs such as ResNet-18/50, MobileNetV2, and RegNetX.

# 2 RELATED WORK

Quantization Li et al. (2023); Sun et al. (2022); Qin et al. (2025); Cai et al. (2020); Harma et al. (2025); Li et al. (2025a; 2024); Zhou et al. (2025); Saxena et al. (2025) is a widely used model compression technique for deep neural networks (DNNs) that reduces model size and accelerates inference by representing

weights and activations with lower bit precision. In practice, two paradigms exist: quantization-aware training (QAT), which incorporates quantization during model re-training (achieving high accuracy but at the cost of additional training on full datasets), and post-training quantization (PTQ), which converts a pre-trained model to low-bit format using only a small unlabeled calibration set without full re-training. While QAT preserves accuracy better, PTQ is efficient in development.

**Post-training Quantization (PTQ)**    Banner et al. (2019); Liu et al. (2023); Nahshan et al. (2021); Banner et al. (2019); Nagel et al. (2020); Wang et al. (2020); Wei et al. (2022); Yuan et al. (2022); Lin et al. (2021); Ding et al. (2022); Lee et al. (2024); Shi et al. (2025); Ding et al. (2025); Zhong et al. (2025); Gong et al. (2025); Wu et al. (2025); Chen et al. (2025); Shen et al. (2025) is the process of determining quantization scale factors and zero-point without retraining or fine-tuning a model's weights. The key challenge in PTQ is estimating the minimum and maximum values of weights and activations, and identifying outliers to clip. Early works on low-bit post-training quantization employed analytic methods to derive optimal clipping thresholds. Banner et al. Banner et al. (2019) proposed limiting activation ranges by statistically deriving activation distributions of tensors and determining the per-channel bit-width. Recent PTQ research has introduced methods to minimize accuracy loss by optimizing layer-wise or block-wise reconstructions. AdaRound Nagel et al. (2020) is a layer-wise reconstruction method that minimizes the local loss by adapting scale factors. To cover cross-layer interaction, BRECQ Li et al. (2021) extends layer-wise reconstruction to a group of layers (blocks) with second-order error approximations. PD-Quant Liu et al. (2023) proposes reconstructing layers and blocks via global loss minimization by comparing the network outputs before and after quantization. To improve the stability of PTQ, QDrop Wei et al. (2022) randomly drops activation quantization during calibration to improve generalization and robustness. Some other works adopt different PTQ formulations. Mr.BiQ Jeon et al. (2022) introduces a non-uniform, multi-level binary quantizer, where both scaling factors and binary codes are treated as learnable parameters and optimized jointly to minimize block-wise reconstruction error. Prepositive Feature Quantization (PFQ) Chu et al. (2024) reorganizes the PTQ framework by moving feature quantization before, rather than after, each layer. In practice, PFQ requires different calibration schedules to effectively align quantized and FP representations in LQ settings.

**Quantization-Aware Training (QAT)**    Hubara et al. (2018); Tailor et al. (2021); Zafrir et al. (2019); Chen et al. (2024); Mishchenko et al. (2019); Wei et al. (2025) integrates quantization into the training process to simulate low-bit computations during forward and backward passes He et al. (2024). Early QAT works Esser et al. (2020); Zhang et al. (2018) rely on the straight-through estimator (STE) Bengio et al. (2013) to approximate gradients of non-differentiable quantization functions, allowing end-to-end optimization. Recent works combined QAT with knowledge distillation Kim et al. (2019) and mixed-precision strategies Wang et al. (2019) to further reduce accuracy degradation, enabling robust deployment of convolutional and Transformer models under aggressive quantization constraints.

## 3   METHOD

Our post-training procedure optimizes all quantization parameters exclusively via the relative entropy (KL divergence) between the full-precision (FP) and quantized output distributions. Let $z_{FP}(x)$ and $z_{LQ}(x)$ denote the FP and quantized logits for input $x$. With temperature $T$ and $p$ as the probability of the model output, define

$$p_{FP}(x) := \text{softmax}(z_{FP}(x)/T), \qquad p_{LQ}(x; \Theta) := \text{softmax}(z_{LQ}(x; \Theta)/T).$$

We determine the scale-factor for each tensor by minimizing the output-level KL divergence on a small calibration set $\mathcal{X}$:

$$\mathcal{L}_{\text{KL\_out}} = \frac{1}{|\mathcal{X}|} \sum_{x \in \mathcal{X}} \text{KL}\big(p_{FP}(x) \,\|\, p_{LQ}(x)\big) = \frac{1}{|\mathcal{X}|} \sum_{x \in \mathcal{X}} \sum_{c} p_{FP}^{(c)}(x) \log \frac{p_{FP}^{(c)}(x)}{p_{LQ}^{(c)}(x)}. \tag{1}$$

To avoid overfitting and preserve hardware-friendly ranges, we add a lightweight parameter regularizer:

$$\mathcal{L}_{\text{reg}} = \|\delta - \delta_0\|_2^2 + \|z - z_0\|_2^2, \tag{2}$$

where $(\delta_0, z_0)$ are the initial (calibration) estimates. The objective is to minimize $\mathcal{L}_{\text{CAT}}$ where,

$$\mathcal{L}_{\text{CAT}} = \mathcal{L}_{\text{KL\_out}} + \lambda_{\text{p}} \mathcal{L}_{\text{reg}}. \tag{3}$$

This procedure ensures that each block is reconstructed to preserve the information content of its FP counterpart, reducing distributional mismatch layer by layer. The refined quantization parameters are then fixed for the remainder of the process, after which we apply logit-level transformation using CAT.

## 3.1 CLUSTER-BASED AFFINE TRANSFORMATION FOR ERROR REDUCTION

After refining quantization parameters, we address residual mismatches directly in the logit space through CAT (Figure 1). To make clustering more effective and computationally efficient, we use a *principal component analysis* (PCA) Wold et al. (1987) to decompose the LQ logits $z_{LQ}$ . PCA reduces the dimensionality of the logits by discarding low-variance components, thereby (i) removing unnecessary complexity, (ii) improving cluster separability, and (iii) lowering clustering cost.

**Cluster-based estimation of affine transformation parameters.** Given calibration logits $\{z_{LQ}(x), z_{FP}(x)\}_{x \in \mathcal{X}}$, we first apply PCA to reduce the dimensionality of $\{z_{LQ}(x)\}$ before clustering. Note that PCA is only used to obtain clusters. The affine transformation itself is always applied in the original logit space of dimension $d$. A clustering model (e.g., $k$-means) is then fitted to the reduced features, yielding cluster assignments $c(x) \in \{1, \ldots, K\}$. For each cluster $C_k = \{x : c(x) = k\}$, we seek Affine Parameters (AP$_K$) $(\gamma_k, \beta_k) \in \mathbb{R}^d$ (with $d$ the dimensionality of logits) that best map quantized logits to their FP counterparts:

$$z_{FP}(x) \approx \gamma_k \odot z_{LQ}(x) + \beta_k, \quad x \in C_k. \tag{4}$$

Where $\odot$ is the element-wise (Hadamard) product. This can be derived in closed form by matching first- and second-order statistics of the two distributions:

$$\mu_{LQ,k} = \frac{1}{|C_k|} \sum_{x \in C_k} z_{LQ}(x), \qquad \mu_{FP,k} = \frac{1}{|C_k|} \sum_{x \in C_k} z_{FP}(x). \tag{5}$$

Here, $\mu_{LQ,k}$ and $\mu_{FP,k}$ denote the mean logits of the LQ and FP models over cluster $C_k$, respectively.

$$\sigma_{LQ,k}^2 = \frac{1}{|C_k|} \sum_{x \in C_k} \left(z_{LQ}(x) - \mu_{LQ,k}\right)^{\odot 2}. \tag{6}$$

$\sigma_{LQ,k}^2$ denotes the variance of LQ logits within cluster $C_k$.

$$\text{cov}_{LQ,FP,k} = \frac{1}{|C_k|} \sum_{x \in C_k} \left(z_{LQ}(x) - \mu_{LQ,k}\right) \odot \left(z_{FP}(x) - \mu_{FP,k}\right). \tag{7}$$

$\text{cov}_{LQ,FP,k}$ denotes the element-wise covariance between LQ and FP logits over cluster $C_k$. The element-wise affine parameters are then estimated as

$$\gamma_k = \frac{\text{cov}_{LQ,FP,k}}{\sigma_{LQ,k}^2 + \epsilon}, \qquad \beta_k = \mu_{FP,k} - \gamma_k \odot \mu_{LQ,k}, \tag{8}$$

where division is elementwise and $\epsilon > 0$ avoids division by zero. Thus, for each cluster $k$, the transformation is obtained by aligning the cluster-wise mean and variance of quantized logits to those of FP, without requiring gradient optimization.

*No-gradient fitting.* We do not require backpropagation through the network to find $(\gamma_k, \beta_k)$. For each cluster $k$, we estimate $\gamma_k$ and $\beta_k$ as parameters to minimize the error between $p_{LQ}$ and $p_{FP}$ using Eq. (4). This black-box procedure fits the affine terms using only a small sample set, avoids gradient computations entirely, and naturally yields low-precision $\{\gamma_k\}_{k=1}^K$ and $\{\beta_k\}_{k=1}^K$ suitable for deployment.

**Calibration, fitting, and inference.** Our two-stage pipeline first optimizes equation 3 to refine quantization parameters $(\delta, z,$ and optional rounding which is defined Appendix A) using only the output-level KL loss. We then find $\gamma$ and $\beta$ through the CAT fitting phase. In inference, CAT requires only a single affine transformation per cluster assignment, introducing negligible overhead while substantially reducing the FP/LQ gap. We first assign the logits $z_{LQ}$ to a cluster $C_k$. We then apply the following $\alpha$-blended correction:

$$\tilde{z} = (1 - \alpha)\,z_{\text{LQ}} \;+\; \alpha\left(\gamma_k \odot z_{\text{LQ}} + \beta_k\right),$$

where $\alpha \in [0, 1]$ controls the contribution of the original quantized output and the affine transformed one. The pseudocode for CAT fitting and inference is detailed in Appendix B.

## 4 EXPERIMENTS

**Setup.** We evaluate CAT on ImageNet-1K Russakovsky et al. (2015) using ResNet-18/50 He et al. (2016), MobileNetV2 Sandler et al. (2018), RegNetX-600MF/3.2GF Radosavovic et al. (2020), and MNasX2 Tan et al. (2019) under different LQ settings, denoted as {weight bit-width}$A$\{activation bit-width}: W4A4, W2A4, W4A2, and W2A2. CAT is compared against strong PTQ baselines (ACIQ-Mix, LAPQ, Bit-Split, AdaRound, QDrop, PD-Quant) with identical hyperparameters to ensure fairness. Models are quantized channel-wise, calibrated with AdaRound (20k iterations, batch size 64, 1,024 samples), and follow prior work by keeping the last layer at 8-bit. CAT adopts the same calibration as PD-Quant, with KL temperature 0.4 and learning rate $4\times10^{-5}$. All experiments are run on NVIDIA L40 GPUs, repeated with three seeds, and we report the mean and standard deviation of Top-1 accuracy. We ablate CAT hyperparameters ($\alpha$, number of clusters (# Clusters), PCA dimension, and clustering samples) to analyze their effect on performance.

**Quantitative Comparison against State-of-the-Art** CAT provides advantages across all quantization settings and architectures (Table 1); however, the magnitude of improvements varies depending on the bit-width configuration and the network's capacity. Overall, our results show that CAT provides the greatest benefits for networks with a larger gap between FP and LQ, such as MNasX2 (W2A2). In such settings, quantization errors accumulate heavily, and CAT's cluster-specific affine correction substantially restores performance. The most extreme case, with both weights and activations quantized to 2 bits (W2A2), reveals the clearest improvements. On ResNet-50, CAT achieves 58.08%, an improvement of +1.05% over PD-Quant. On MNasX2, CAT reaches 29.20%, outperforming PD-Quant by +1.25%. Gains are also consistent across ResNet-18 (+0.32%), MobileNetV2 (+0.51%), and RegNetX-3.2GF (+1.06%). Similarly, with 4-bit weights and 2-bit activations (W4A2), where the activation bottleneck is severe, CAT consistently delivers improvements. On ResNet-18, CAT yields 58.68%, exceeding PD-Quant by +0.11%, while on MNasX2, CAT improves accuracy to 40.14%, a substantial +0.71% increase. Gains are also observed on MobileNetV2 (+0.39%) and RegNetX-600MF (+0.35%). These findings demonstrate that W2A2 and W4A2 are the most error-prone regimes, and CAT's targeted corrections are especially effective at resolving their distorted feature representations.

In another asymmetric setting, W2A4, CAT also provides consistent improvements. For instance, CAT improves over PD-Quant by +0.19% on ResNet-18, +0.45% on ResNet-50, and +0.35% on RegNetX-3.2GF.

Table 2: Top-1 accuracy (%) on ImageNet-1K for various PTQ methods across architectures.

| Methods | Bits (W/A) | ResNet-18 | ResNet-50 | MobileNetV2 | RegNetX-600MF | RegNetX-3.2GF | MNasX2 |
|---|---|---|---|---|---|---|---|
| Full Prec. | 32/32 | 71.01 | 76.63 | 72.62 | 73.52 | 78.46 | 76.52 |
| ACIQ-Mix Banner et al. (2019) | | 67.00 | 73.80 | – | – | – | – |
| LAPQ Nahshan et al. (2021) | | 60.30 | 70.00 | 49.70 | 57.71 | 55.89 | 65.32 |
| Bit-Split Wang et al. (2020) | 4/4 | 67.56 | 73.71 | – | – | – | – |
| AdaRound Nagel et al. (2020) | | 67.96 | 73.88 | 61.52 | 68.20 | 73.85 | 68.86 |
| QDrop Wei et al. (2022) | | 69.17 | 75.15 | 68.07 | 70.91 | 76.40 | 72.81 |
| PD-Quant Liu et al. (2023) | | $69.14 \pm 0.10$ | $75.07 \pm 0.09$ | $68.18 \pm 0.02$ | $70.96 \pm 0.03$ | $76.54 \pm 0.02$ | $73.24 \pm 0.02$ |
| **Ours** | | $\mathbf{69.18 \pm 0.09}$ | $\mathbf{75.12 \pm 0.07}$ | $\mathbf{68.22 \pm 0.01}$ | $\mathbf{70.98 \pm 0.01}$ | $\mathbf{76.61 \pm 0.02}$ | $\mathbf{73.31 \pm 0.05}$ |
| LAPQ | | 0.18 | 0.14 | 0.13 | 0.17 | 0.12 | 0.18 |
| Adaround | | 0.11 | 0.12 | 0.15 | – | – | – |
| QDrop | 2/4 | 64.57 | 70.09 | 53.37 | 63.18 | 71.96 | 63.23 |
| PD-Quant | | $65.10 \pm 0.02$ | $70.84 \pm 0.06$ | $55.30 \pm 0.24$ | $63.92 \pm 0.24$ | $72.36 \pm 0.13$ | $63.32 \pm 0.24$ |
| **Ours** | | $\mathbf{65.26 \pm 0.06}$ | $\mathbf{71.29 \pm 0.02}$ | $\mathbf{55.47 \pm 0.22}$ | $\mathbf{64.2 \pm 0.28}$ | $\mathbf{72.71 \pm 0.12}$ | $\mathbf{63.96 \pm 0.32}$ |
| QDrop | | 57.56 | 63.26 | 17.30 | 49.73 | 62.79 | 34.12 |
| PD-Quant | 4/2 | $58.57 \pm 0.18$ | $64.24 \pm 0.02$ | $20.14 \pm 0.52$ | $51.17 \pm 0.27$ | $62.68 \pm 0.08$ | $39.43 \pm 0.34$ |
| **Ours** | | $\mathbf{58.68 \pm 0.15}$ | $\mathbf{64.38 \pm 0.06}$ | $\mathbf{20.53 \pm 0.53}$ | $\mathbf{51.52 \pm 0.26}$ | $\mathbf{63.03 \pm 0.05}$ | $\mathbf{40.14 \pm 0.27}$ |
| QDrop | | 51.42 | 55.45 | 10.28 | 39.01 | 54.38 | 23.59 |
| PD-Quant | 2/2 | $52.87 \pm 0.03$ | $57.03 \pm 0.12$ | $13.65 \pm 0.63$ | $40.71 \pm 0.13$ | $55.08 \pm 0.13$ | $27.95 \pm 0.69$ |
| **Ours** | | $\mathbf{53.19 \pm 0.07}$ | $\mathbf{58.08 \pm 0.11}$ | $\mathbf{14.16 \pm 0.61}$ | $\mathbf{41.38 \pm 0.1}$ | $\mathbf{56.14 \pm 0.09}$ | $\mathbf{29.20 \pm 0.76}$ |

Note: PD-Quant results are reproduced from our runs.

On MobileNetV2, CAT achieves $55.47\%$, improving upon PD-Quant by $+0.17\%$, while MNasX2 benefits from a $+0.64\%$ gain. Interestingly, W2A4 tends to produce more diverse outcomes compared to W2A2 or W4A2: while low-bit weights distort the learned filters and increase the risk of incorrect feature extraction, the higher activation precision (4-bit) preserves a broader dynamic range, enabling richer but more variable behavior than the severely compressed 2-bit activation cases. At higher precision (W4A4), where the discrepancy between full-precision (FP) and low-bit (LQ) networks is relatively small, CAT provides only marginal improvements since its corrections are designed to bridge larger gaps. In this regime, CAT achieves accuracy on par with or slightly better than prior methods. On ResNet-18, CAT obtains $69.18\%$, essentially matching PD-Quant ($69.14\%$), while on RegNetX-3.2GF, CAT improves to $76.61\%$, the best among all compared approaches. Similar trends are observed on MobileNetV2 ($68.22\%$) and ResNet-50 ($75.12\%$), confirming that when quantization noise is less severe, CAT closely mimics the FP network but cannot deliver substantial gains. An important observation in this is that networks with a larger discrepancy between FP and LQ outcomes benefit the most from CAT in LQ settings. This effect arises because low-capacity models have limited redundancy to absorb quantization errors, making CAT's correction particularly impactful. Across all architectures and bit-widths, CAT either matches or outperforms prior state-of-the-art PTQ methods. The improvements are most pronounced under LQ settings (W2A2, W4A2, W2A4) and on compact models, where quantization errors are most destructive. CAT achieves more effective reduction than plain affine mapping, thereby establishing a new state-of-the-art in PTQ. Appendix I discusses a statistical analysis of CAT performance across the cross of model parameters and LQ settings.

**Enhance PTQ methods with CAT**    Table 3 (More comprehensive results are provided in Appendix Table 5) provides a direct comparison of multiple PTQ baselines with and without the proposed CAT correction across different architectures and bit-width settings. The values in parentheses indicate the performance difference $\Delta = (\text{CAT} - \text{Base})$, where green arrows ($\uparrow$) mark improvements and red arrows ($\downarrow$) indicate degradations. Across nearly all architectures and methods, CAT consistently enhances performance. The improvements are particularly pronounced in LQ settings (W2A2 and W4A2), where quantization errors are most severe. For example, CAT boosts PD-Quant on ResNet-50 (W2A2) by $+1.21\%$, on RegNetX-3.2GF (W2A2) by $+1.01\%$, and on MNasX2 (W2A2) by $+1.24\%$. Similarly, for W4A2, CAT improves QDrop on MNasX2 by more than $+1\%$ and PD-Quant on MobileNetV2 by $+0.37\%$. These gains confirm that CAT is most effective in highly error-prone regimes, where its cluster-aware affine correction recovers distorted

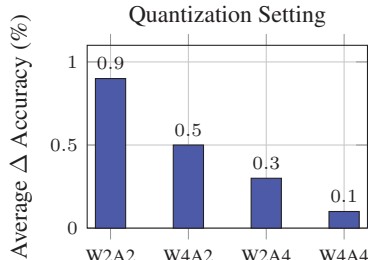

Figure 2: Average improvements $\Delta = (\text{CAT} - \text{Base})$ across quantization settings.

Table 3: ResNet-18 Top-1 Accuracy (%) with/without CAT. $\Delta$ shows improvement (green ↑) or degradation (red ↓).

| | **W4A4** | | | | **W4A2** | | | |
|---|---|---|---|---|---|---|---|---|
| | Adaround | BRECQ | QDrop | PD-Quant | Adaround | BRECQ | QDrop | PD-Quant |
| Base | 2.59 | 69.32 | 69.17 | 69.14 | 1.18 | 49.44 | 57.91 | 58.57 |
| + CAT | 2.58 | 69.35 | 69.24 | 69.18 | 1.14 | 50.49 | 58.12 | 58.68 |
| $\Delta$ | -0.01 ↓ | +0.03 ↑ | +0.07 ↑ | +0.04 ↑ | -0.04 ↓ | +1.05 ↑ | +0.21 ↑ | +0.11 ↑ |
| | **W2A4** | | | | **W2A2** | | | |
| | Adaround | BRECQ | QDrop | PD-Quant | Adaround | BRECQ | QDrop | PD-Quant |
| Base | 18.48 | 62.69 | 64.52 | 65.10 | 2.59 | 40.78 | 51.47 | 52.86 |
| + CAT | 22.04 | 62.67 | 64.76 | 65.35 | 2.58 | 41.75 | 51.76 | 53.19 |
| $\Delta$ | +3.56 ↑ | -0.02 ↓ | +0.24 ↑ | +0.25 ↑ | -0.01 ↓ | +0.97 ↑ | +0.29 ↑ | +0.32 ↑ |

representations. In asymmetric quantization (W2A4), CAT again provides steady improvements, albeit with smaller margins (e.g., ResNet-50 +0.42%, RegNetX-3.2GF +0.45%), while in higher-precision settings (W4A4), the effect is marginal but consistently non-negative, reflecting that the baseline already closely approximates full-precision. An important observation is that smaller-capacity architectures exhibit disproportionately large gains from CAT, providing new evidence that networks with a large FP–LQ gap benefit the most in LQ settings. For instance, LQ MNasX2 gains more than +1%, demonstrating that CAT is particularly valuable for compact models that lack redundancy to absorb quantization errors. Overall, Table 3 highlights CAT's robustness as a drop-in enhancement: it supplements diverse PTQ baselines (Adaround, BRECQ, Qdrop, PD-Quant) consistently yielding improved accuracy while never severely degrading performance. This consistency demonstrates CAT's generality and compatibility with existing PTQ pipelines. Figure 2 shows the average improvement of CAT over the Base method. The largest gain is observed for W2A2 quantization (0.9%) across all PTQ methods and models. The second-highest improvement occurs for W4A2 (0.5%), highlighting the effectiveness of CAT under very low-bit activation quantization. For W2A4 and W4A4, the accuracy increases by 0.3% and 0.1%, respectively, demonstrating the generalization capability of CAT. Notably, 2-bit activation settings benefit the most from CAT, since their severely limited precision reduces the ability to preserve information entropy, making them more reliant on CAT's error reduction mechanism. Appendix H represents LQ ViT error reduction using CAT.

## 5 ABLATION STUDY

**Ablation on Blending Coefficient $\alpha$.** This study ablates the effect of the blending coefficient $\alpha$, where $\alpha = 0$ corresponds to using only PTQ logits without any CAT correction, and $\alpha = 1$ corresponds to relying entirely on CAT-corrected logits without involving the original PTQ. As shown in Fig. 3, varying $\alpha$ directly influences the final accuracy across different bit-width regimes (The comprehensive ablation of Blending Coefficient $\alpha$ for different LQ settings and architectures provided in Appendix D). For the 2-bit activation quantization (W2A2 and W4A2), moderate blending ($\alpha \approx 0.3$–$0.4$) consistently provides the highest performance, indicating that CAT is most effective when used as a supplement to the baseline quantized outputs rather than a full replacement. In the asymmetric regime (W2A4), accuracy is relatively stable across a broad range of $\alpha$, reflecting that higher-precision activations reduce sensitivity

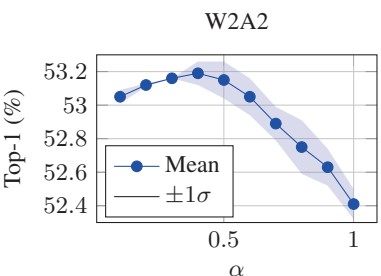

Figure 3: Top-1 accuracy for W2A2 with mean and $\pm 1\sigma$ range.

to blending, although small gains are still observed around $\alpha \approx 0.4$.

At higher precision (W4A4), the performance is nearly flat across all values of $\alpha$, as the quantized model is already close to full-precision accuracy. Overall, these results demonstrate that CAT reliably improves PTQ performance, especially in LQ settings, but that using CAT alone ($\alpha = 1$) is suboptimal.

**Ablation on the Number of Clusters.** We next ablate the influence of the number of clusters used in CAT, as shown in Fig. 5 for ResNet-18 W2A4 with PCA dimension fixed to $50$ and blending coefficient $\alpha = 0.6$(The comprehensive ablation of the number of clusters for different LQ settings and architectures is provided in Appendix E). For W2A2, performance is highest with a small number of clusters and gradually declines as the cluster count increases. This suggests that in extremely quantized regimes, a compact cluster partition provides stable corrections, while too many clusters lead to overfitting of noise in the heavily distorted logit space. For W4A2, a similar trend is observed: accuracy peaks at very low cluster counts (1–8 clusters) and then decreases slowly with more clusters, indicating diminishing returns once the coarse logit structure has been captured. In the asymmetric case W2A4, accuracy remains largely stable across a broad range of cluster

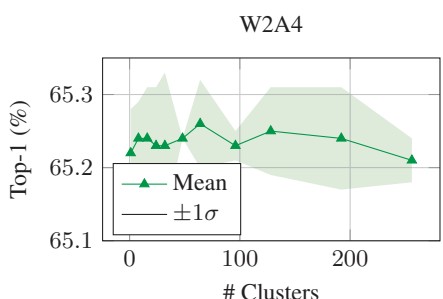

Figure 4: Top-1 accuracy for W2A4 across different # Clusters (mean $\pm 1\sigma$).

counts, confirming that higher-precision activations mitigate sensitivity to clustering granularity. At higher precision (W4A4), accuracy is nearly unaffected by # clusters, as the quantized logits already closely approximate the full-precision distribution. Overall, these results demonstrate that the optimal number of clusters is inherently fitted to the nature of quantization precision: lower precision reduces diversity in the logit space and thus favors fewer clusters, whereas higher precision allows richer structures that can benefit from larger cluster counts. This further suggests that clustering is not only a useful but also an essential component of the affine transformation, enabling it to adaptively restore quantization errors according to the underlying representation capacity of the quantized network.

**Ablation on PCA Dimension $k$.** We further study the effect of the PCA dimension $k$ used to reduce the logit space before clustering, as shown in Fig. 5 for ResNet-18 under the W2A4 quantization setting (The comprehensive ablation of the PCA dimension for different LQ settings and architectures is provided in Appendix F). For W2A2, performance is maximized when $k$ is very small (1–5) and gradually decreases as $k$ increases. This indicates that in extremely quantized networks, only the coarse structure of the logits can be reliably captured, and projecting onto a compact subspace avoids fitting noise. For W4A2, a similar but weaker trend is observed: accuracy peaks when $k$ is kept small ($\leq 10$) and remains stable for moderate values before slightly degrading with very large $k$. In the asymmetric W2A4 setting, accuracy is largely stable across the full range of $k$, suggesting that higher activation precision preserves sufficient feature variability to tolerate richer subspaces

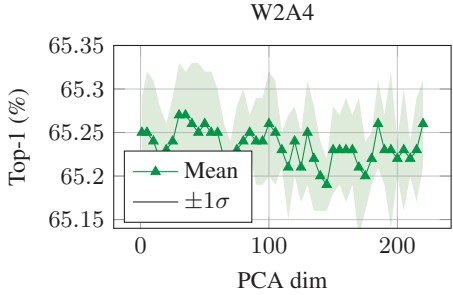

Figure 5: Top-1 accuracy for W2A4 across different PCA dim (mean $\pm 1\sigma$).

without significant overfitting. At higher precision (W4A4), accuracy is almost invariant to the PCA dimension, as the quantized logits already closely match the full-precision distribution and PCA reduction plays only a minor role. Results indicate that the optimal PCA dimension depends on the severity of quantization: lower-precision networks benefit from aggressive dimensionality reduction that filters out noise, while

higher-precision networks can afford larger $k$ without degradation. This highlights PCA as an essential regularization step that adapts the clustering space to the effective diversity of the quantized representations.

**Ablation on CAT fitting sample size.** Table 4 studies the impact of the number of samples for the fitting of CAT on ResNet-18 on the W2A2 setting (The comprehensive ablation of sample size on CAT's performance for different ResNet-18 LQ settings is provided in Appendix G). We observe a consistent monotonic trend with diminishing returns: accuracy improves rapidly when increasing samples from 10 to 500–1000, and then plateaus. Under W2A2, the mean Top-1 rises from $46.48\%$ (10 samples) to $53.04\%$ (1,000), a gain of $\approx +6.6$ points, with only marginal changes beyond 1,000 (e.g., $53.15\%$ at 100,000). W4A2 shows a similar but smaller effect ($53.67\% \to 58.54\%$, $\approx +4.9$). In contrast, higher-precision settings are less sensitive: W2A4 improves by $\approx +2.7$ ($62.60\% \to 65.33\%$), while W4A4

Table 4: Top-1 accuracy for W2A2 across different numbers of samples CAT fitting.

| # Samples | Top-1 (%) |
|-----------|-----------|
| 10        | 46.48     |
| 50        | 51.79     |
| 100       | 52.49     |
| 500       | 53.02     |
| 1K        | 53.04     |
| 10K       | 53.15     |
| 100K      | 53.14     |

gains only $\approx +1.1$ ($68.13\% \to 69.26\%$). The shaded $\pm 1\sigma$ bands narrow as sample size grows, indicating more stable calibration and reduced run-to-run variability, especially pronounced in W2A2/W4A2. A calibration set of $\sim$500–1,000 samples captures nearly all attainable gains across regimes, striking a favorable accuracy/cost trade-off. Using $\geq$ 10k samples yields negligible additional improvements, particularly in W4A4, where performance is near its ceiling.

## 6 LIMITATIONS

While CAT consistently improves accuracy over PTQ baselines, it introduces additional parameters due to the clustering step and the cluster-specific affine corrections. In particular, PTQ baselines do not maintain any auxiliary parameters beyond the quantized model itself, whereas CAT requires storing the clustering model and the affine coefficients $(\gamma, \beta)$ for each cluster. Here, we analyse additional parameters by considering the K-Means clustering algorithm with $k$ means. This overhead grows linearly with the number of clusters $k$ and the logit dimensionality $d$ (equal to the number of classes). The additional parameter count of CAT can therefore be approximated as $\text{CAT}_{\#\text{params}} = (k \times d \text{ for cluster-wise affine coefficients}) + (k \times d \text{ for } k\text{-means centroids})$. For example, for ResNet-18 with 11.6 million parameters, the number of model parameters in addition to CAT with $k = 50$ and $d = 1000$ is $11,600,000 + 2 * (50 * 1000) = 11,700,000$. In this example, CAT adds only $\sim 0.9\%$ parameter overhead relative to the baseline model. Nevertheless, this extra storage may be undesirable for extremely resource-constrained deployments, which we identify as a limitation of CAT compared to PTQ baselines.

## 7 CONCLUSION

We studied the gap between full-precision (FP) and low-bit quantized (LQ) networks, focusing on restoring quantization errors through affine transformations. Our initial findings showed that a plain affine mapping with a uniform parameter set can even worsen PTQ performance. To overcome this, we introduced Cluster-based Affine Transformation (CAT), which leverages the clusterability of quantized logits and applies cluster-specific affine corrections. CAT consistently improves accuracy under low-bit settings with only negligible parameter overhead, achieving state-of-the-art performance on ImageNet-1K. These results highlight that cluster-aware error reduction is a powerful and generalizable strategy for enhancing PTQ, particularly in extremely low-bit regimes.

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
