# APPENDIX FOR CAT

## A   QUANTIZATION NOTATION

To better ground the discussion of CAT, we start by defining the notation used in neural network quantization. Quantization is the process of compacting a neural network by reducing the precision of full-precision weights and activations, considering the given bit-width allowance. For notational simplicity, we consider the quantization of a scalar value. Let $f \in \mathbb{R}$ denote a floating-point value (weight or activation) with real range

$$f_{\min} = \min(f), \qquad f_{\max} = \max(f).$$

For a given bit-width $b$, the representable integer range is defined as

$$q_{\min} = -2^{b-1}, \quad q_{\max} = 2^{b-1} - 1,$$

where $q_{\min}$ and $q_{\max}$ denote the minimum and maximum values of the quantized representation. Assuming *scale factor* $\delta$ and *zero-point* $z$ as parameters to convert to convert floating-point value $f$ to its quantized value $q$ with $N$-bit representation, then we have,

$$\delta = \frac{f_{\max} - f_{\min}}{q_{\max} - q_{\min}}, \qquad z = \left\lfloor q_{\min} - \frac{f_{\min}}{\delta} \right\rceil.$$

The quantization and dequantization mappings are then given by

$$q = \text{clip}\left( \left\lfloor \frac{f}{\delta} \right\rceil + z, \ q_{\min}, \ q_{\max} \right),$$

$$\hat{f} = \delta \cdot (q - z),$$

$\hat{f}$ is the reconstructed quantized value, and clip($\cdot$) ensures $q \in [q_{\min}, q_{\max}]$. Weights quantization is simpler since their values remain constant during PTQ, whereas activations quantization requires estimating their minimum and maximum. In post-training quantization (PTQ), $f_{\min}$ and $f_{\max}$ are estimated from a small calibration dataset. Different strategies include *min–max calibration*, which directly uses observed extrema, *percentile or clipping methods*, which discard outliers, and *optimization-based approaches* such as AdaRound or BRECQ, which refine $\delta$ (and sometimes $z$) by minimizing reconstruction error between FP and quantized outputs. We denote a quantization configuration by $WbAb$, where $Wb$ and $Ab$ specify the bit-widths used for weights and activations, respectively (e.g., $W2A4$ indicates 2-bit weights with 4-bit activations).

## B  CLUSTER-BASED AFFINE TRANSFORMATION (CAT) ALGORITHM

---

**Algorithm 1** CAT Fitting

---

**Require:** all_q $\in \mathbb{R}^{N \times C}$, all_fp $\in \mathbb{R}^{N \times C}$, $Clusters$, pca_dim
**Ensure:** CLUSTER-MODEL, PCA, $\{\gamma_c\}_{c=1}^{K}$, $\{\beta_c\}_{c=1}^{K}$
 1: $q\_features \leftarrow$ PCA.$decomposition$(pca_dim, all_q)
 2: cluster_ids $\leftarrow$ CLUSTER-MODEL.$fit(Clusters, q\_features)$
 3: **for** $cid \leftarrow 1 \rightarrow Clusters$ **do**
 4:     idxs $\leftarrow \{i \mid$ cluster_ids$[i] = cid\}$
 5:     **if** $|$idxs$| = 0$ **then**
 6:         $\gamma\_cid \leftarrow \mathbf{1}_C$; $\beta\_cid \leftarrow \mathbf{0}_C$
 7:         **continue**
 8:     **end if**
 9:     $Q \leftarrow$ all_q[idxs], $\quad F \leftarrow$ all_fp[idxs]
10:     $\mu_Q \leftarrow$ Mean$(Q)$, $\quad \mu_F \leftarrow$ Mean$(F)$
11:     $\sigma_Q^2 \leftarrow \max($Var$(Q), 10^{-8})$
12:     $\gamma_{cid} \leftarrow \dfrac{\text{Mean}[(Q - \mu_Q) \odot (F - \mu_F)]}{\sigma_Q^2}$
13:     $\beta_{cid} \leftarrow \mu_F - \gamma_{cid} \odot \mu_Q$
14: **end for**
15: **return** CLUSTER-MODEL, $\{\gamma_c\}, \{\beta_c\}$

---

---

**Algorithm 2** CAT Inferencing

---

**Require:** $q\_logits$, CLUSTER-MODEL, $\{\gamma_c\}, \{\beta_c\}, \alpha$
**Ensure:** corrected logits, PCA
 1: $q\_feature \leftarrow$ PCA.$decomposition(q\_np)$
 2: $cluster\_ids \leftarrow$ CLUSTER-MODEL.$predict(q\_feature)$
 3: $corrected \leftarrow [\,]$
 4: **for** $i = 1 \rightarrow |q\_logits|$ **do**
 5:     $q \leftarrow q\_logits[i]$
 6:     $cid \leftarrow cluster\_ids[i]$
 7:     $\gamma \leftarrow \gamma\_cid, \beta \leftarrow \beta\_cid$
 8:     $affine\_corrected \leftarrow q \cdot \gamma + \beta$
 9:     $blended \leftarrow q + \alpha \cdot (affine\_corrected - q)$
10:     Append $blended$ to $corrected$
11: **end for**
12: **return** $corrected$

---

## C    ENHANCE PTQ METHODS WITH CAT

Table 5: Comparison of Quantization Methods (with/without CAT) across Architectures on ImageNet-1K. Values in parentheses show $\Delta = (\text{CAT} - \text{Base})$. Green ↑: CAT improves. Red ↓: CAT worse. The comprehensive description of thís table's results is provided in Section 4

| Bits (W/A) | Methods | ResNet-18 | ResNet-50 | MobileNetV2 | RegNetX-600MF | RegNetX-3.2GF | MNasX2 |
|---|---|---|---|---|---|---|---|
| 4/4 | Adaround
+ CAT ($\Delta$) | 2.59
2.58 (-0.01 ↓) | 1.57
1.57 (0.00) | 24.22
28.51 (+4.29 ↑) | –
 | 0.95
0.98 (+0.03 ↑) | 0.68
0.75 (+0.07 ↑) |
| 4/4 | BRECQ
+ CAT ($\Delta$) | 69.32
69.35 (+0.03 ↑) | 74.45
74.48 (+0.03 ↑) | 4.56
6.48 (+1.92 ↑) | 69.67
69.72 (+0.05 ↑) | 74.57
74.68 (+0.11 ↑) | 66.26
66.57 (+0.31 ↑) |
| 4/4 | Qdrop
+ CAT ($\Delta$) | 69.17
69.24 (+0.07 ↑) | 75.08
75.18 (+0.10 ↑) | 67.86
68.01 (+0.15 ↑) | 70.78
70.89 (+0.11 ↑) | 76.31
76.46 (+0.15 ↑) | 72.91
73.04 (+0.13 ↑) |
| 4/4 | PD-Quant
+ CAT ($\Delta$) | 69.17
69.27 (+0.10 ↑) | –
– | 68.20
68.28 (+0.08 ↑) | 70.9
71.08 (+0.18 ↑) | –
– | –
– |
| 4/2 | Adaround
+ CAT ($\Delta$) | 1.18
1.14 (-0.04 ↓) | –
– | 24.22
28.51 (+4.29 ↑) | –
– | –
– | 0.15
0.17 (+0.02 ↑) |
| 4/2 | BRECQ
+ CAT ($\Delta$) | 49.44
50.49 (+1.05 ↑) | 36.59
38.47 (+1.88 ↑) | 0.23
0.43 (+0.20 ↑) | 6.79
11.22 (+4.43 ↑) | 6.48
10.16 (+3.68 ↑) | 0.82
1.73 (+0.91 ↑) |
| 4/2 | Qdrop
+ CAT ($\Delta$) | 57.91
58.12 (+0.21 ↑) | 63.15
63.54 (+0.39 ↑) | 16.56
16.93 (+0.37 ↑) | 49.83
50.05 (+0.22 ↑) | 62.15
62.70 (+0.55 ↑) | 30.43
31.44 (+1.01 ↑) |
| 4/2 | PD-Quant
+ CAT ($\Delta$) | 58.66
58.80 (+0.14 ↑) | 64.26
64.49 (+0.23 ↑) | 19.84
20.21 (+0.37 ↑) | 51.11
51.52 (+0.41 ↑) | 62.63
63.05 (+0.42 ↑) | 39.59
40.32 (+0.73 ↑) |
| 2/4 | Adaround
+ CAT ($\Delta$) | 18.48
22.04 (+3.56 ↑) | –
– | 0.36
0.74 (+0.38 ↑) | –
– | –
– | 0.00
0.00 (0.00) |
| 2/4 | BRECQ
+ CAT ($\Delta$) | 62.69
62.67 (-0.02 ↓) | 64.23
64.44 (+0.21 ↑) | 0.12
0.20 (+0.08 ↑) | 51.57
51.65 (+0.08 ↑) | 64.39
64.57 (+0.18 ↑) | 0.23
0.32 (+0.09 ↑) |
| 2/4 | Qdrop
+ CAT ($\Delta$) | 64.52
64.76 (+0.24 ↑) | 69.95
70.50 (+0.55 ↑) | 54.10
54.41 (+0.31 ↑) | 63.17
63.50 (+0.33 ↑) | 71.59
72.11 (+0.52 ↑) | 63.39
64.09 (+0.70 ↑) |
| 2/4 | PD-Quant
+ CAT ($\Delta$) | 65.10
65.35 (+0.25 ↑) | 70.87
71.29 (+0.42 ↑) | 55.58
55.72 (+0.14 ↑) | 63.90
64.31 (+0.41 ↑) | 72.21
72.66 (+0.45 ↑) | 63.06
63.82 (+0.76 ↑) |
| 2/2 | Adaround
+ CAT ($\Delta$) | 2.59
2.58 (-0.01 ↓) | –
– | –
– | –
– | –
– | –
– |
| 2/2 | BRECQ
+ CAT ($\Delta$) | 40.78
41.75 (+0.97 ↑) | 13.78
16.69 (+2.91 ↑) | 0.10
0.16 (+0.06 ↑) | 0.68
2.03 (+1.35 ↑) | 1.73
3.45 (+1.72 ↑) | 0.12
0.18 (+0.06 ↑) |
| 2/2 | Qdrop
+ CAT ($\Delta$) | 51.47
51.76 (+0.29 ↑) | 55.46
56.81 (+1.35 ↑) | 11.81
12.25 (+0.44 ↑) | 38.92
39.75 (+0.83 ↑) | 54.12
55.26 (+1.14 ↑) | 23.04
24.11 (+1.07 ↑) |
| 2/2 | PD-Quant
+ CAT ($\Delta$) | 52.86
53.18 (+0.32 ↑) | 57.12
58.33 (+1.21 ↑) | 13.92
14.33 (+0.41 ↑) | 40.67
41.32 (+0.65 ↑) | 55.23
56.24 (+1.01 ↑) | 28.09
29.33 (+1.24 ↑) |

# D    ABLATION OF $\alpha$ FOR DIFFERENT ARCHITECTURES

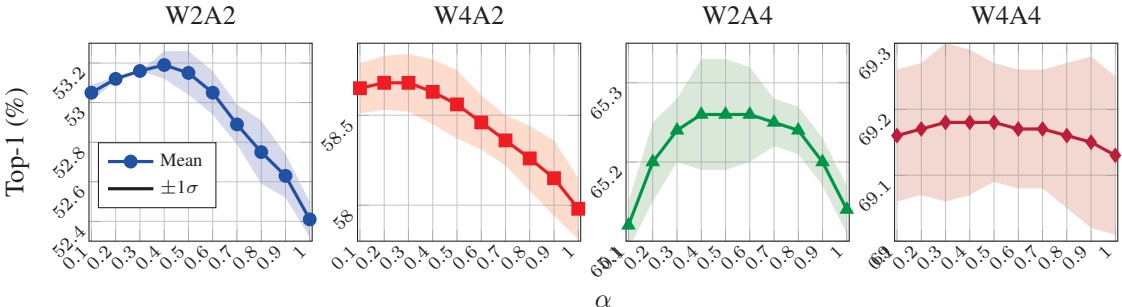

Figure 6: Ablation of $\alpha$ for ResNet-18 under different quantization precisions (W2A2, W2A4, W4A2, W4A4). The solid line denotes the mean Top-1 accuracy across three runs, and the shaded region indicates $\pm 1\sigma$. The comprehensive description of this ablation is provided in Section 5

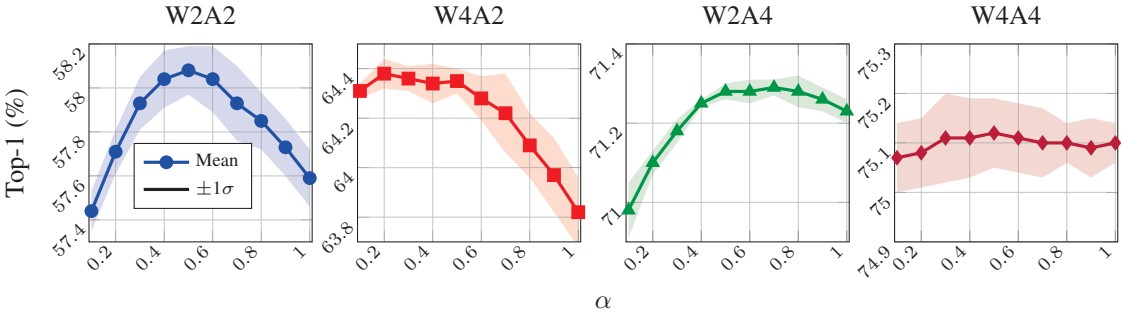

Figure 7: Ablation of $\alpha$ for ResNet-50 under different quantization precisions (W2A2, W2A4, W4A2, W4A4). The solid line denotes the mean Top-1 accuracy across three runs, and the shaded region indicates $\pm 1\sigma$. Across quantization settings, $\alpha$ controls the balance between the base quantized output and CAT correction. Performance peaks at moderate $\alpha$ ($\approx 0.3$–$0.5$) for W2A2 and W4A2, remains stable for W2A4, and is nearly flat for W4A4, indicating that CAT provides the most benefit in severely quantized regimes with low-bit activations.

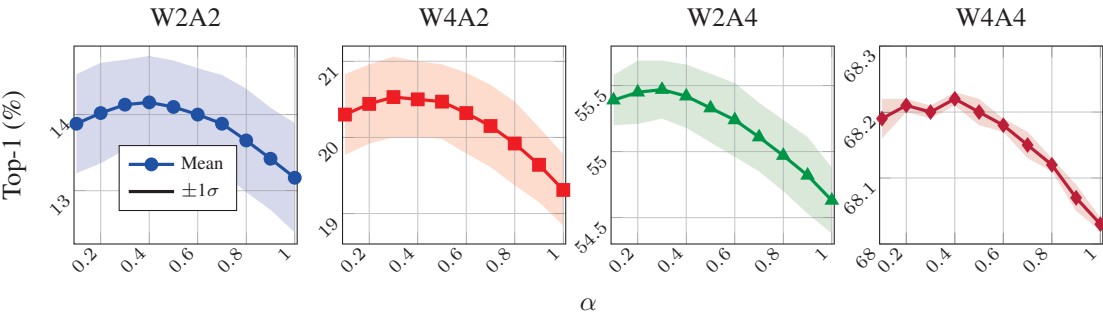

Figure 8: Ablation of $\alpha$ for MobileNetV2 under different quantization precisions (W2A2, W2A4, W4A2, W4A4). The solid line denotes the mean Top-1 accuracy across three runs, and the shaded region indicates $\pm 1\sigma$. For MobileNetV2, CAT gains are largest under severe activation quantization: W2A2 and W4A2 peak at moderate $\alpha$ ($\approx 0.3$–$0.4$) before dropping sharply, while W2A4 shows only mild variation and W4A4 remains essentially flat, indicating that $\alpha$ is most critical when activations are 2-bit.

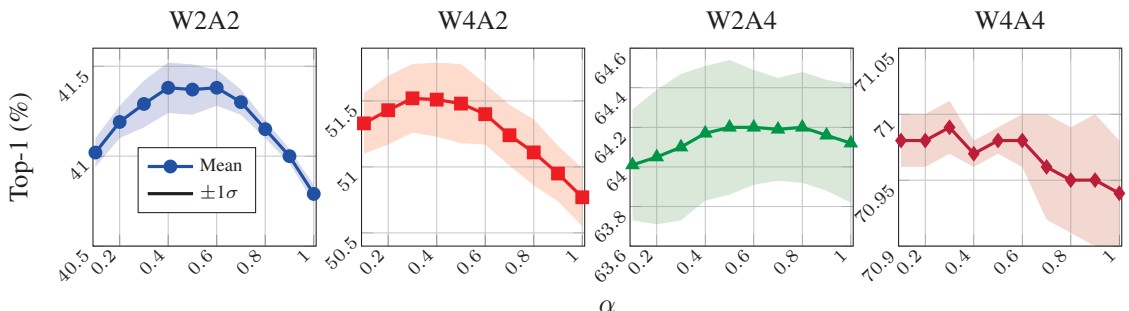

Figure 9: Ablation of $\alpha$ for RegNetX-600M under different quantization precisions (W2A2, W2A4, W4A2, W4A4). The solid line denotes the mean Top-1 accuracy across three runs, and the shaded region indicates $\pm 1\sigma$. For RegNetX-600M, $\alpha$ is most influential in 2-bit activation regimes: W2A2 and W4A2 peak at moderate $\alpha$ ($\approx 0.3$–$0.5$) and decline at larger values, whereas W2A4 shows only mild variation and W4A4 remains flat around 71%.

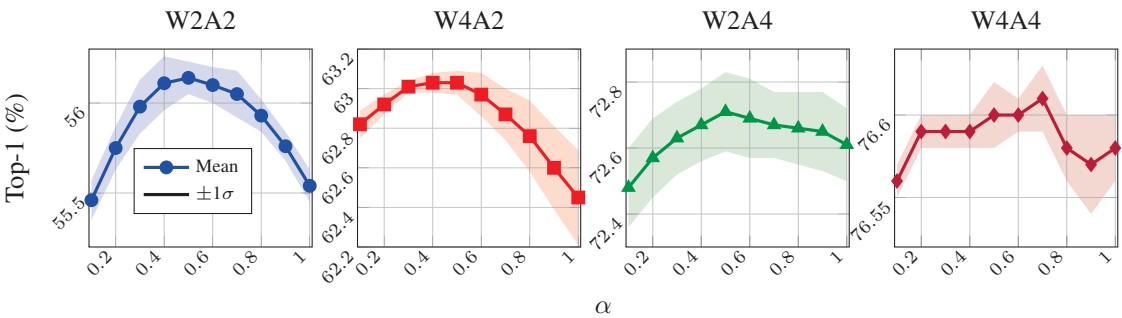

Figure 10: Ablation of $\alpha$ for RegNetX-3.2G under different quantization precisions (W2A2, W2A4, W4A2, W4A4). The solid line denotes the mean Top-1 accuracy across three runs, and the shaded region indicates $\pm 1\sigma$. For RegNetX-3.2G, $\alpha$ tuning is most beneficial in 2-bit activation regimes: W2A2 and W4A2 peak at moderate values ($\alpha \approx 0.3$–$0.5$) before declining, while W2A4 shows only mild variation and W4A4 remains flat near 76.6%.

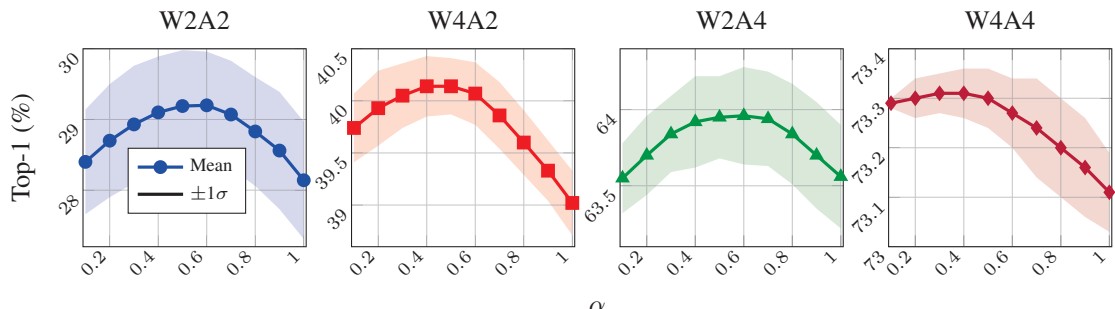

Figure 11: Ablation of $\alpha$ for MNasX2 under different quantization precisions (W2A2, W2A4, W4A2, W4A4). The solid line denotes the mean Top-1 accuracy across three runs, and the shaded region indicates $\pm 1\sigma$. For MNasX2, $\alpha$ tuning is most beneficial in 2-bit activation regimes: W2A2 and W4A2 peak at moderate values ($\alpha \approx 0.3$–$0.5$) before degrading, while W2A4 shows only mild variation and W4A4 remains flat around 73.3%.

# E ABLATION OF THE NUMBER OF CLUSTERS FOR DIFFERENT ARCHITECTURES

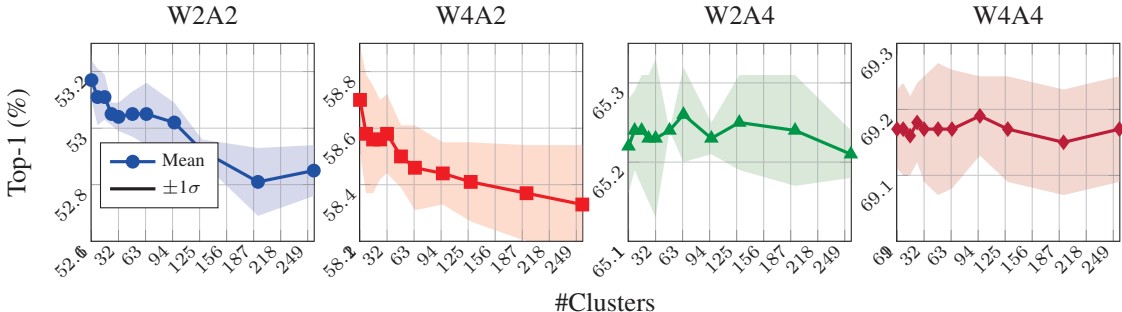

Figure 12: Ablation of the number of clusters for ResNet-18 (PCA=50, $\alpha = 0.6$) under different quantization precisions (W2A2, W4A2, W2A4; W4A4 pending). Solid line is the mean Top-1 over three runs; shaded region is $\pm 1\sigma$. The comprehensive description of this ablation is provided in Section 5

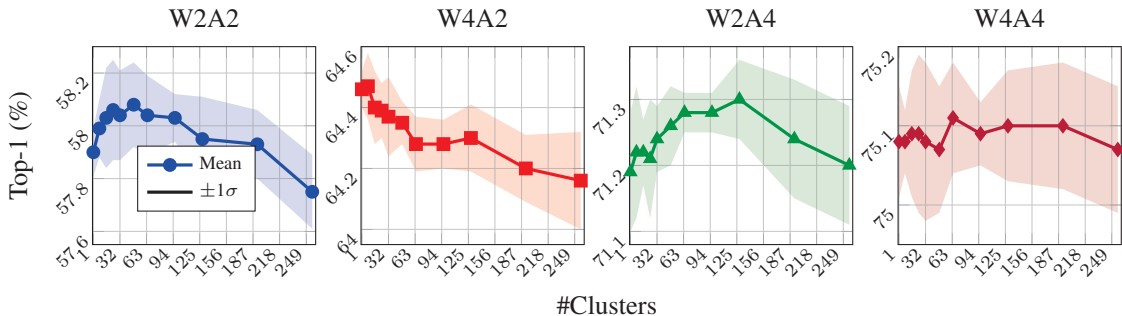

Figure 13: Ablation of number of clusters for ResNet-50 under different quantization precisions (W2A2, W2A4, W4A2, W4A4). The solid line denotes the mean Top-1 accuracy across three runs, and the shaded region indicates $\pm 1\sigma$. On ResNet-50, W2A2 benefits slightly from moderate $K$ (peak ~58.08% near $K \approx 48$), whereas W4A2 declines with larger $K$, W2A4 shows a shallow peak near $K \approx 128$, and W4A4 is flat. Small-to-moderate $K$ suffices; very large $K$ offers no consistent gains.

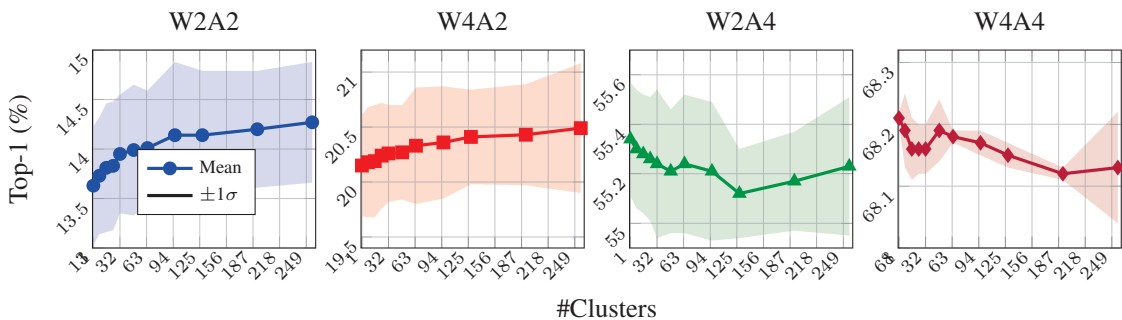

Figure 14: Ablation of number of clusters for MobileNetV2 under different quantization precisions (W2A2, W2A4, W4A2, W4A4). The solid line denotes the mean Top-1 accuracy across three runs, and the shaded region indicates $\pm 1\sigma$. On MobileNetV2, W2A2 and W4A2 improve steadily with larger $K$, peaking at $K=256$, while W2A4 declines slightly, and W4A4 is flat. Thus, more clusters help only in severe activation quantization.

## F    ABLATION OF PCA DIMENSIONS FOR DIFFERENT ARCHITECTURES

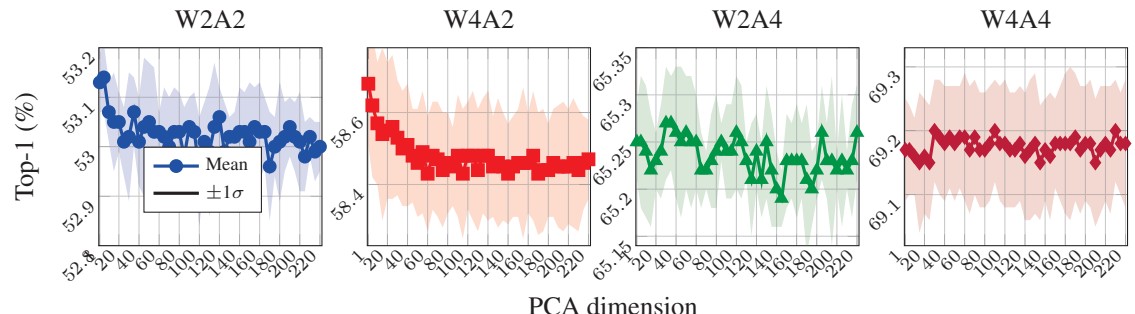

Figure 15: Ablation of PCA dimension $k$ for ResNet-18 under different quantization precisions (W2A2, W4A2, W2A4, W4A4). Solid lines show mean Top-1 accuracy over three runs; the shaded region is $\pm 1\sigma$. The comprehensive description of this ablation is provided in Section 5

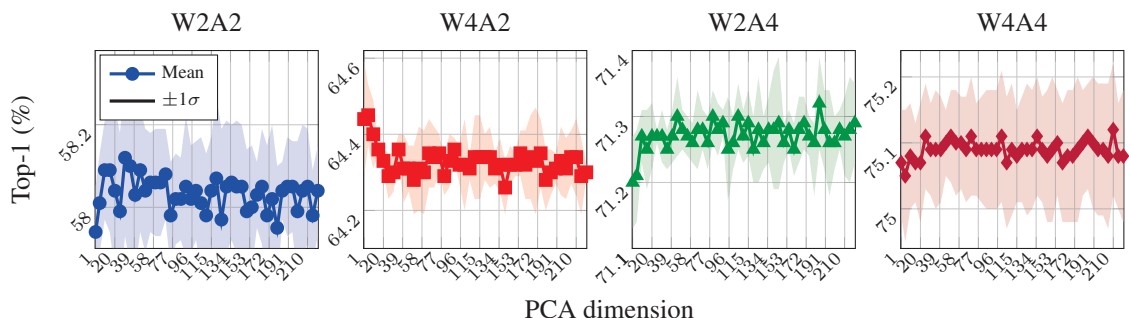

Figure 16: Ablation of PCA dimension $d$ for ResNet-50 under different quantization precisions (W2A2, W2A4, W4A2, W4A4). The solid line is the mean Top-1 across three runs; the shaded band shows $\pm 1\sigma$. For ResNet-50, the number of clusters $K$ controls how finely CAT partitions the logit space. Under 2-bit activations, performance improves rapidly as $K$ increases from small values and peaks at a moderate range ($K \approx 8$–$32$), after which it plateaus or slightly declines due to over-partitioning and noisier parameter estimates. In W2A2, Top-1 rises steadily up to mid-range $K$ and then saturates; W4A2 follows a similar pattern with a slightly earlier plateau. By contrast, W2A4 shows only mild gains across $K$, while W4A4 is essentially flat, reflecting the smaller FP–LQ gap in higher-precision regimes. Overall, these trends indicate that a moderate cluster count strikes the best balance between expressivity and robustness, especially when activations are quantized to 2-bit.

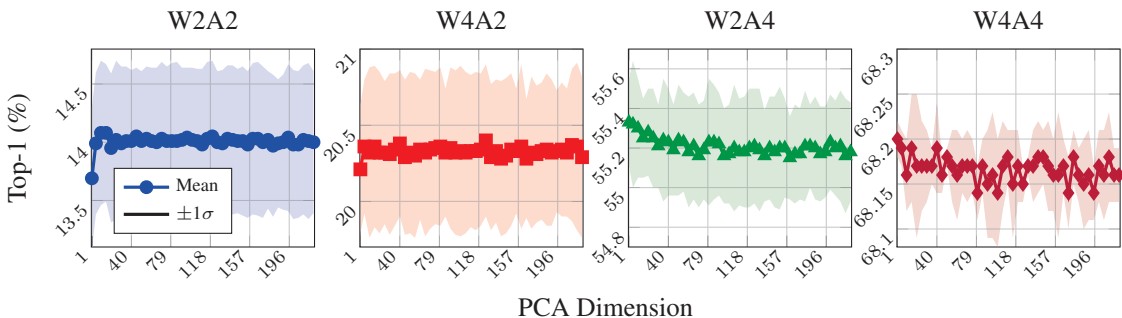

Figure 17: Effect of PCA dimension on **MobileNetV2** Top-1 accuracy under different quantization settings (W2A2, W4A2, W2A4, W4A4). The solid line is the mean over three runs; the shaded area is $\pm 1\sigma$. On MobileNetV2, PCA dimension matters mainly with 2-bit activations: W2A2 peaks at small $p$ ($\sim$10–15) and then plateaus, while W4A2/W2A4 change little and W4A4 is flat. A compact subspace ($p\approx$10–40) offers a good robustness–accuracy trade-off.

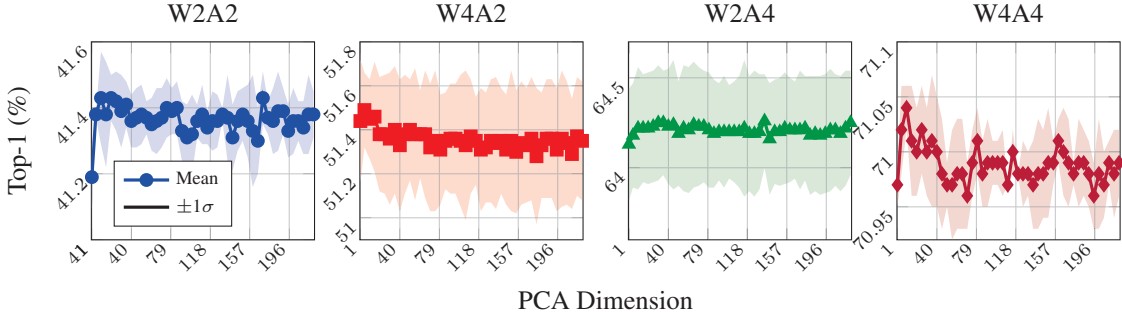

Figure 18: Effect of PCA dimension on **RegNetX-600MF** Top-1 accuracy under different quantization settings (W2A2, W4A2, W2A4, W4A4). The solid line is the mean over three runs; the shaded area is $\pm 1\sigma$. On RegNetX-600MF, PCA dimension matters only in W2A2, which peaks around $p$=10–20 and then plateaus; W4A2, W2A4, and W4A4 remain essentially flat. Thus, small subspaces suffice for this model.

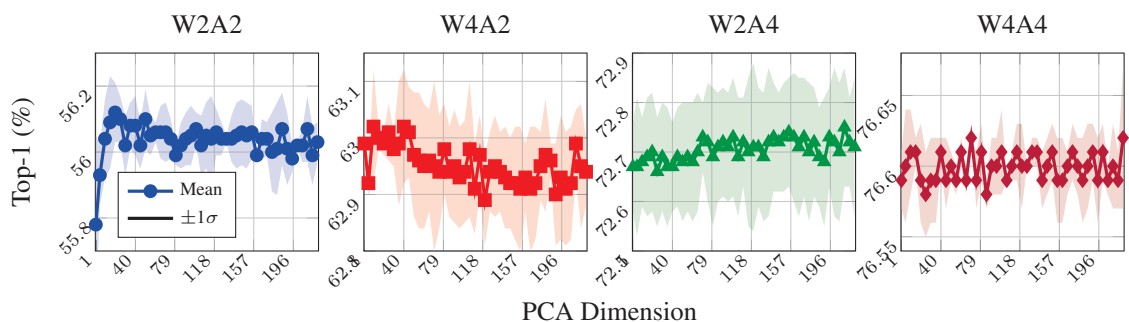

Figure 19: Effect of PCA dimension on **RegNetX-3.2GF** Top-1 accuracy under different quantization settings (W2A2, W4A2, W2A4, W4A4). The solid line is the mean over three runs; the shaded area is $\pm 1\sigma$. On RegNetX-3.2GF, W2A2 benefits slightly from small $p$ (peaking near $p\approx 20$), while W4A2, W2A4, and W4A4 are essentially flat. A compact PCA subspace ($p\approx 10$–40) suffices for robust performance.

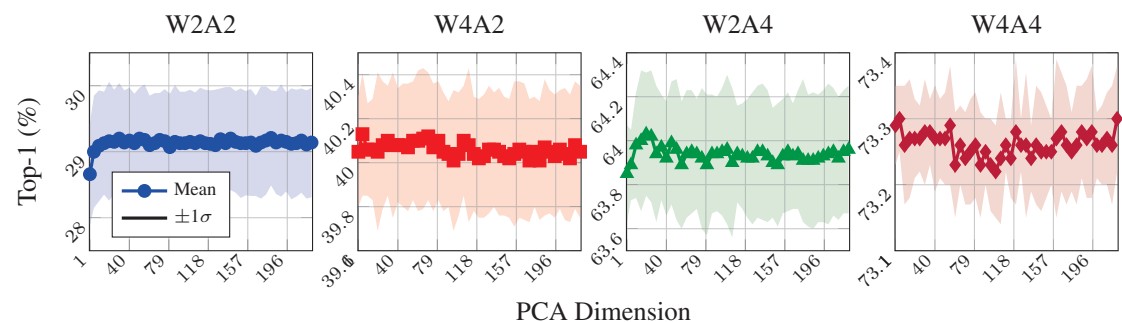

Figure 20: Effect of PCA dimension on **MNasX2** Top-1 accuracy under different quantization settings (W2A2, W4A2, W2A4, W4A4). The solid line is the mean over three runs; the shaded area is $\pm 1\sigma$. On MNasX2, W2A2 gains slightly up to $p\approx 30$ and then plateaus; W4A2, W2A4, and W4A4 are essentially flat. Thus, a compact PCA subspace ($p\approx 10$–40) suffices.

# G ABLATION ON CAT FITTING SAMPLE SIZE

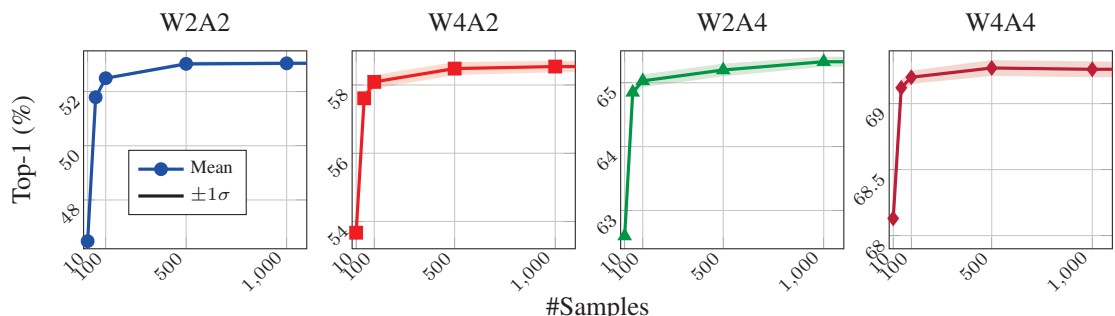

Figure 21: Ablation on the number of samples for ResNet-18 under different quantization settings (W2A2, W4A2, W2A4, W4A4). Solid lines show mean Top-1 accuracy; the shaded region is $\pm1\sigma$. For ResNet-18, increasing the number of fitting samples yields clear gains that saturate once a few hundred to a thousand samples are used. In W2A2, Top-1 rises sharply from 46.5% at 10 samples to 53.0–53.1% by 500–1,000 samples, with only marginal changes thereafter (53.14–53.15% at 10k–100k). W4A2 follows the same trend, improving from 53.7% (10) to 58.5% (1k–10k) with diminishing returns beyond 500 samples. In W2A4, accuracy increases from 62.6% (10) to 65.3% (1k) and flattens around 65.2–65.3%. W4A4 similarly moves from 68.1% (10) to 69.3% (0.5k–10k) and then saturates. Overall, CAT achieves most of its benefit with ~500–1,000 samples, indicating that modest data budgets suffice for effective post-training error restoration.

## H   CAT IMPROVEMENT ON VIT QUANTIZATION VIA DIFFERENT PTQ METHODS

Table 6 reports Top-1 accuracy for ViT, Swin, and DeiT models under W2A2 and W4A4 quantization with BRECQ and QDrop. CAT consistently improves performance across all settings. The gains are most notable in ultra-low precision (W2A2), with improvements up to +0.3%–0.4%, while W4A4 shows smaller but steady benefits, confirming CAT's generalizability across architectures and quantization methods.   Table 7

Table 6: Top-1 accuracy (%) for BRECQ and QDrop with/without CAT under W2A2 and W4A4.

| Method | Variant | W2A2 | | | | | | W4A4 | | | | | |
|--------|---------|-------|-------|--------|--------|--------|--------|-------|-------|--------|--------|--------|--------|
| | | ViT-S | ViT-B | Swin-S | Swin-B | DeiT-S | DeiT-B | ViT-S | ViT-B | Swin-S | Swin-B | DeiT-S | DeiT-B |
| **BRECQ** | Base | 0.63 | 2.042 | 3.43 | 4.1 | 4.84 | 12.46 | 73.96 | 79.23 | 81.07 | 82.6 | 69.82 | 77.64 |
| | + CAT | 0.75 | 2.17 | 3.52 | 4.27 | 5.04 | 12.79 | 74.09 | 79.42 | 81.11 | 82.65 | 70.09 | 77.7 |
| **QDrop** | Base | 3.36 | 4.77 | 12.85 | 15.69 | 4.1 | 30.53 | 69.51 | 81.96 | 80.79 | 82.25 | 69.93 | 78.01 |
| | + CAT | 3.34 | 4.83 | 12.83 | 15.77 | 4.27 | 30.72 | 69.54 | 82 | 80.86 | 82.30 | 70.13 | 78.05 |

Table 7: Top-1 accuracy (%) for BRECQ and QDrop with/without CAT on DeiT-S and DeiT-B under different precisions.

| Method | Variant | W2A2 | | W4A4 | | W2A6 | | W4A6 | | W6A6 | |
|--------|---------|--------|--------|--------|--------|--------|--------|--------|--------|--------|--------|
| | | DeiT-S | DeiT-B | DeiT-S | DeiT-B | DeiT-S | DeiT-B | DeiT-S | DeiT-B | DeiT-S | DeiT-B |
| **BRECQ** | Base | 4.84 | 12.46 | 69.82 | 77.64 | 65.25 | 75.22 | 77.64 | 80.83 | 76.7 | 80.49 |
| | + CAT | 5.04 | 12.79 | 70.09 | 77.70 | 65.30 | 75.28 | 77.65 | 80.9 | 76.87 | 80.5 |
| **QDrop** | Base | 4.10 | 30.53 | 69.93 | 78.01 | 65.71 | 76.0 | 77.9 | 80.9 | 77.74 | 80.82 |
| | + CAT | 4.27 | 30.72 | 70.13 | 78.05 | 66.04 | 76.11 | 77.95 | 80.94 | 77.82 | 80.86 |

shows that CAT consistently enhances accuracy for both DeiT-S and DeiT-B across all tested precisions. The improvements are most pronounced under very low-bit precision, reaching up to +0.4% at W2A2 (e.g., DeiT-B with BRECQ: 12.79% vs. 12.46%) and +0.3% at W2A6 (e.g., DeiT-S with QDrop: 66.04% vs. 65.71%). At more moderate bit-widths (W4A4, W4A6, W6A6), CAT provides smaller but consistent gains in the range of +0.1–0.2%, indicating that the benefit decreases as the quantized model approaches full-precision accuracy. Importantly, improvements are observed for both BRECQ and QDrop baselines and across both DeiT-S and DeiT-B models, highlighting CAT's robustness and generality as a lightweight correction mechanism for transformer architectures under different quantization regimes.

## I   DISCUSSION

Our experiments demonstrate that CAT consistently improves the performance of a wide range of PTQ baselines across architectures and precision settings. On average, we observe absolute accuracy gains of +1.56% for W2A2, +1.19% for W4A2, +0.39% for W2A4, and +0.10% for W4A4 within the largest parameter regime (20–40M). Smaller-capacity models (3–7M) also benefit, though to a lesser extent, with gains of +0.68% (W2A2) and +0.96% (W4A2). These trends confirm that CAT is particularly effective in challenging low-precision regimes, while improvements diminish as precision increases. A key observation is that lower activation precision yields the largest relative improvements. In both W2A2 and W4A2 regimes, CAT provides the strongest corrections, while W2A4 and W4A4 exhibit smaller yet consistent gains. This pattern indicates that activation quantization is the dominant factor driving representational degradation, and CAT's cluster-specific corrections directly mitigate this bottleneck. In contrast, weight precision plays a

comparatively minor role: lowering weights to 2-bit (while keeping activations higher) causes less severe degradation and thus smaller relative improvements with CAT.

We also find that larger models (20–40M parameters, e.g., ResNet-50, RegNetX-3.2GF) exhibit the highest absolute improvements in LQ settings, surpassing $+1\%$ in both W2A2 and W4A2. Our interpretation is that these models possess greater capacity to process complex information, but when activations are severely quantized, their representational potential is limited by activation precision. CAT effectively restores this information flow by aligning low-precision activations with their full-precision distributions. In contrast, compact models (3–7M, e.g., MobileNetV2, MNasX2) still benefit from CAT but show smaller relative gains, as their limited redundancy constrains the extent to which CAT can compensate. In summary, the results highlight three consistent trends: (i) CAT is most effective when activation precision is low, (ii) larger models benefit more in absolute terms, and (iii) weight quantization alone has a weaker influence on CAT's efficacy compared to activation quantization. Together, these findings position CAT as a practical and robust enhancement for PTQ pipelines, particularly in ultra-low-precision and high-capacity scenarios where quantization errors are most detrimental.

Table 8: Average Top-1 improvement $\Delta$ (%) of CAT over Base by parameter regimes and precision. Regimes: 3–7M (MobileNetV2, MNasX2, RegNetX-600MF), 10–20M (ResNet-18), 20–40M (ResNet-50, RegNetX-3.2GF). Darker green indicates larger improvement.

| Params Regime | W2A2 | W4A2 | W2A4 | W4A4 |
|---|---|---|---|---|
| $< 10M$ | 0.68 | 0.96 | 0.32 | 0.37 |
| $10M - 20M$ | 0.53 | 0.47 | 0.16 | 0.07 |
| $20M - 40M$ | 1.56 | 1.19 | 0.39 | 0.10 |