# OpenReview forum: "CAT: Post-Training Quantization Error Reduction via Cluster-based Affine Transformation"
_ICLR.cc/2026/Conference — ICLR 2026 Conference Withdrawn Submission_

### Official Review · Reviewer_Tbsw · 2025-10-24

**Soundness:** 2
**Presentation:** 2
**Contribution:** 2
**Rating:** 2
**Confidence:** 5

**Summary:**

This paper proposes Cluster-based Affine Transformation (CAT), a framework to reduce PTQ errors in deep neural networks. Instead of applying a single global affine correction, CAT performs cluster-specific affine mappings between quantized and full-precision logits. Each cluster’s scale and bias parameters are estimated in closed form without gradient updates, using a small calibration set. Experiments on ImageNet-1K with diverse architectures show that CAT consistently improves accuracy—especially under extreme low-bit settings (e.g., W2A2)—while adding negligible parameter overhead.

**Strengths:**

1. Using cluster-specific affine corrections to align quantized and full-precision outputs is intuitive, lightweight, and easy to implement.
2. CAT consistently improves Top-1 accuracy across multiple architectures (ResNet, MobileNetV2, RegNetX, MNasX2) and bit-widths, especially in 2-bit settings.
3. The paper includes detailed ablation studies (α blending, cluster number, PCA dimension, sample size) demonstrating robustness and stability.

**Weaknesses:**

1. The paper does not provide any analysis of the inference-time overhead introduced by CAT. Although PCA and cluster assignment are claimed to be lightweight, they may still incur noticeable latency. A detailed runtime and latency evaluation is necessary to substantiate the claim of negligible cost.
2. While CAT consistently improves accuracy, the absolute improvements are relatively small (often below 1%) on several benchmarks. The practical significance of these gains, given the added complexity, should be discussed more thoroughly.
3. Results on how CAT generalizes to ViTs and LLMs would make the work more convincing and broadly relevant.
4. The paper lacks deeper discussion or comparison with other recent PTQ techniques. For example, OBQ, GPTQ, COMQ ...

[1] Frantar, Elias, et al. "Optimal brain compression: A framework for accurate post-training quantization and pruning." Advances in Neural Information Processing Systems 35 (2022): 4475-4488.

[2] Frantar, Elias, et al. "Gptq: Accurate post-training quantization for generative pre-trained transformers." arXiv preprint arXiv:2210.17323 (2022).

[3] Zhang, Aozhong, et al. "Comq: A backpropagation-free algorithm for post-training quantization." IEEE Access (2025).

**Questions:**

1. How does CAT affect inference latency and memory footprint?
2. How are cluster IDs assigned for unseen inputs during inference?

---

### Official Review · Reviewer_WCQU · 2025-10-30

**Soundness:** 2
**Presentation:** 2
**Contribution:** 3
**Rating:** 2
**Confidence:** 4

**Summary:**

The paper introduces a novel method called Cluster-based Affine Transformation (CAT) to reduce the accuracy gap between full-precision (FP) models and their low-bit quantized (LQ) counterparts. The authors observe that traditional plain affine transformations, which apply a single uniform correction, are ineffective or even detrimental in aggressive low-bit (e.g., 2-bit or 4-bit) Post-Training Quantization (PTQ) settings

**Strengths:**

1. The method is well-motivated. It directly targets the failure of plain affine transformation in low-bit PTQ, a claim supported by the authors in Table 1.
2. The authors test CAT on a diverse set of CNN architectures (ResNet-18/50, MobileNetV2, RegNet, MNasX2) and multiple low-bit settings (W4A4, W2A4, W4A2, W2A2).
3. The paper includes a comprehensive set of ablation studies (Section 5) that validate the key hyperparameters introduced by CAT, including the blending factor ($\alpha$), the number of clusters, the PCA dimension, and the calibration sample size.

**Weaknesses:**

1. The paper discusses parameter overhead but provides no analysis of the computational (latency) cost introduced by the CAT-specific steps during inference (PCA projection, cluster assignment, and affine transformation).
2. The paper is functionally incomplete. It makes at least nine distinct references to appendices for crucial details, including pseudocode, comprehensive results, additional ablations, and results for Vision Transformers (ViT). Without these, the paper is not fully verifiable.
3. The citation style is disruptive, often using long, uncontextualized lists of references and creating confusing attributions.
4. The main method section only vaguely suggests "k-means" , leaving the reader to infer from the limitations section  that it was the algorithm used. No justification for this choice over other clustering algorithms is provided.
5. All experiments were run with only three seeds. This is a very small sample size to confidently claim statistical significance for the many marginal improvements reported.

**Questions:**

1. Missing Appendices: The paper references Appendices A through I, which are not included. Could you please provide these appendices? They seem to contain essential information, including pseudocode, full experimental results, and (critically) the results for Vision Transformers (ViT).
2. Inference Latency: The "Limitations" section analyzes parameter overhead  but omits computational cost. What is the total inference latency overhead (in ms or FLOPS) introduced by CAT, including the PCA projection, cluster assignment, and final affine transformation?
3. Clustering Algorithm: The "Method" section does not specify the clustering algorithm, but Limitations implies K-Means was used. Could you confirm this and justify why K-Means was chosen over other algorithms.
4. Can you report results of your experiments for with at least 10 randomly selected seeds?

---

### Official Review · Reviewer_Q6KF · 2025-10-31

**Soundness:** 3
**Presentation:** 3
**Contribution:** 2
**Rating:** 4
**Confidence:** 3

**Summary:**

The paper proposes "CAT" post-training quantization of neural networks using cluster-based linear transformations. The clusters are found using principal component analysis and then. A subset of image samples is used to find the CAT parameters for PCA projections, cluster assignments, and the transformation parameters. These transformations have to be applied at inference time to obtain network output.

**Strengths:**

The paper is well written and provides a good background on post-training quantization. The experimental results are convincing and show improvements over the SOTA. The proposed method works especially well for extremely low-bit-count models. It also contains multiple ablation studies, which provide a better understanding of what is happening.

**Weaknesses:**

The authors should address the following issue in the paper:

1. A comparison with BRECQ (Block Reconstruction for post-training quantization). The authors have referenced and discussed the paper, but have not included it in the comparison table. One of the results for 2/4 for ResNet-18 is better in the BRECQ paper.
2. The method requires overhead in terms of space, as is listed as one of the weaknesses. What the paper does not mention is the inference time overhead. The PCA and affine transformations will require complex and specialized activations. The authors should compare the inference time of their method to competing methods. Overall, the idea of quantization is to decrease the energy and space utilization of these models. Decreasing the model size but adding additional parameters and creating expensive and complex activations does not lead to a fair comparison.
3. The number of hyperparameters that are introduced and then have to be tuned does not, in my opinion, justify the small increase in accuracy. The work seems incremental, and it would be OK if there were significant accuracy gains, but that is not the case here.

**Questions:**

1. Table comparing the runtime of their method and the SOTA, including BRECQ.
2. How difficult is it to tune the hyperparameters?
3. Can you use other clustering algorithms, and how do they compare in performance? K-means usually does not provide consistent results for me.

---

### Official Review · Reviewer_sBgh · 2025-10-31

**Soundness:** 3
**Presentation:** 2
**Contribution:** 2
**Rating:** 2
**Confidence:** 5

**Summary:**

This paper proposes CAT, a two-stage PTQ framework combining KL-based distillation to refine quantization parameters and cluster-specific affine transformations to correct residual errors. Experiments on vision models and tasks are conducted.

**Strengths:**

1. It is novel to first cluster  and then use affine transformation to correct residual errors.
2. Can be used to improve PTQ methods.

**Weaknesses:**

1. The experiments are limited to relatively small vision models. Could the authors clarify why evaluations were not extended to other modalities (e.g., language or multimodal models)? Since the proposed method is general, it would be valuable to demonstrate its applicability beyond vision tasks.
2. The performance gains appear quite marginal compared to existing quantization methods, particularly in Tables 2 and 3.
3. The method involves fine-tuning quantization scales and zero-points via a distillation-like procedure, which may introduce nontrivial computational cost. An ablation study evaluating the impact of this stage (with vs. without distillation) would help assess its necessity.
4. The recent work PEQA also fine-tunes scales for quantized models. A direct comparison or discussion of the differences would strengthen the paper.

**Questions:**

In Equation (2), is $\delta$ the scale factor and $z$ the zero-point? The symbol $z$ was previously used to denote logits in the definition of $p_{FP}(x)$, so reusing it here is confusing. It would be clearer to use distinct notation for these variables.

---

### Note · Authors · 2025-12-03

**Comment:**

Dear ICLR Program Chairs,

I hope you are doing well. I would like to formally request the withdrawal of our ICLR 2026 submission entitled “CAT: Post-Training Quantization Error Reduction via Cluster-based Affine Transformation.”

After careful discussion with my co-authors, we have decided to continue improving the manuscript by addressing the reviewers’ comments in depth, introducing additional experiments, and strengthening the methodological contributions. We believe that a more complete and mature version of the paper will be better suited for a future submission.

We kindly ask you to proceed with the withdrawal at your earliest convenience.
Thank you very much for your time and for the efforts of the reviewers.

Best regards,
Ali Zoljodi
(on behalf of all co-authors)

**Withdrawal Confirmation:**

I have read and agree with the venue's withdrawal policy on behalf of myself and my co-authors.